# Drug Response of Patient-Derived Lung Cancer Cells Predicts Clinical Outcomes of Targeted Therapy

**DOI:** 10.3390/cancers16040778

**Published:** 2024-02-14

**Authors:** Sunshin Kim, Youngjoo Lee, Bo Ram Song, Hanna Sim, Eun Hye Kang, Mihwa Hwang, Namhee Yu, Sehwa Hong, Charny Park, Beung-Chul Ahn, Eun Jin Lim, Kum Hui Hwang, Seog-Yun Park, Jin-Ho Choi, Geon Kook Lee, Ji-Youn Han

**Affiliations:** 1Research Institute, National Cancer Center, Goyang 10408, Republic of Korea; ksunshin@ncc.re.kr (S.K.); yjlee@ncc.re.kr (Y.L.); 75758@ncc.re.kr (B.R.S.); 2305205@ncc.re.kr (H.S.); sidhdgg789@gmail.com (E.H.K.); polaris0628@ncc.re.kr (M.H.); namheeyu@ncc.re.kr (N.Y.); sehwahong@ncc.re.kr (S.H.); charn78@ncc.re.kr (C.P.); 2Center for Lung Cancer, National Cancer Center, Goyang 10408, Republic of Korea; abcduke@ncc.re.kr (B.-C.A.); dms35@ncc.re.kr (E.J.L.); hkhle@ncc.re.kr (K.H.H.); 13016@ncc.re.kr (J.-H.C.); 3Division of Hematology and Oncology, Department of Internal Medicine, National Cancer Center, Goyang 10408, Republic of Korea; 4Department of Pathology, National Cancer Center, Goyang 10408, Republic of Korea; 11740@ncc.re.kr (S.-Y.P.); gklee@ncc.re.kr (G.K.L.)

**Keywords:** patient-derived cell, targeted therapy, EGFR mutation, ALK fusion, lung cancer, predictive marker

## Abstract

**Simple Summary:**

We introduced a drug screening platform for patient-derived cells (PDCs) from lung cancer patients. The drug sensitivity results of short-term-cultured PDCs were significantly associated with clinical outcomes in patients with targeted therapy. The PDC drug screening platform was useful to predict the clinical outcomes of patients treated with targeted therapy and facilitated precision medicine for patients without relevant driver oncogenes. We expect that the PDC drug screening platform, as a feasible tool, can be used to guide target therapy for lung cancer patients and may overcome the limitation of genomic assay-based targeted therapy.

**Abstract:**

Intratumor heterogeneity leads to different responses to targeted therapies, even within patients whose tumors harbor identical driver oncogenes. This study examined clinical outcomes according to a patient-derived cell (PDC)-based drug sensitivity test in lung cancer patients treated with targeted therapies. From 487 lung cancers, 397 PDCs were established with a success rate of 82%. In 139 PDCs from advanced non-small-cell lung cancer (NSCLC) patients receiving targeted therapies, the standardized area under the curve (AUC) values for the drugs was significantly correlated with their tumor response (*p* = 0.002). Among 59 chemo-naive EGFR/ALK-positive NSCLC patients, the PDC non-responders showed a significantly inferior response rate (RR) and progression-free survival (PFS) for the targeted drugs than the PDC responders (RR, 25% vs. 78%, *p* = 0.011; median PFS, 3.4 months [95% confidence interval (CI), 2.8–4.1] vs. 11.8 months [95% CI, 6.5–17.0], *p* < 0.001). Of 25 EGFR-positive NSCLC patients re-challenged with EGFR inhibitors, the PDC responder showed a higher RR than the PDC non-responder (42% vs. 15%). Four patients with wild-type EGFR or uncommon EGFR-mutant NSCLC were treated with EGFR inhibitors based on their favorable PDC response to EGFR inhibitors, and two patients showed dramatic responses. Therefore, the PDC-based drug sensitivity test results were significantly associated with clinical outcomes in patients with EGFR- or ALK-positive NSCLC. It may be helpful for predicting individual heterogenous clinical outcomes beyond genomic alterations.

## 1. Introduction

Advances in molecular biology and genetic technology have led to substantial progress in personalized anti-cancer treatment [1]. Lung cancer is perhaps the best-known application of personalized treatment, based on the development and use—over the past few decades—of several drugs targeting unique oncogenic driver mutations [2]. For example, an EGFR-targeting drug is the first molecularly targeted therapy that has led to a paradigm shift in the management of patients with advanced non-small-cell lung cancer (NSCLC) [3]. This drug has been used as a standard first-line treatment for patients with lung cancer harboring EGFR exon 19 deletion or exon 21 L858R mutations from multiple randomized clinical trials [4,5,6,7,8]. Molecular testing for patients newly diagnosed with non-small-cell lung cancer (NSCLC), including alterations in the genes *EGFR*, *ALK*, *KRAS*, *ROS1*, *BRAF*, *NTRK*, *MET*, *RET*, and *ERBB2*, is routinely performed to detect genetic alterations and thus determine the appropriate targeted therapy. However, even when genotype-directed targeted therapies are appropriately chosen, several clinical challenges remain. Of these, tumor heterogeneity is a major issue; it can unexpectedly reduce the efficacy of targeted therapy and thus worsen survival outcomes [9,10]. Therefore, multiple efforts are underway to overcome these limitations and enhance the efficacy of targeted therapy in lung cancer. For example, biological in vitro system models, including patient-derived cell cultures (PDCs), patient-derived xenografts, and patient-derived organoids, constitute innovative attempts to better predict drug sensitivity in individual tumors [11,12]. The advantages of patient-derived xenografts and patient-derived organoid models include their abilities to sufficiently recapitulate each tumor’s heterogeneous molecular characteristics. However, neither model is easily implemented in clinical settings because of the limited drug screening scalability, low success rate, and high cost and time requirements [11]. In contrast, the PDC model is a biological system that is easily manipulated and allows for the rapid testing of drug sensitivity [11]. It has been broadly evaluated in various cancer studies, including studies that demonstrate its clinical relevance in terms of predicting a tumor’s drug sensitivity [13,14]. However, prolonged and repetitive culturing can result in a PDC that lacks the heterogenous features of the original tumor. This weakness of the PDC model led to the proposal of a short-term PDC model that can accurately represent the original tumor [15,16,17]. Thus, the present study evaluated the clinical feasibility and efficiency of a large-scale drug screening platform based on a short-term PDC model; the findings are expected to provide guidance for lung cancer treatment.

## 2. Materials and Methods

### 2.1. PDC Establishment

Tumor samples were collected from patients diagnosed with advanced lung cancer at the National Cancer Center Hospital (Goyang, Republic of Korea) from August 2016 to June 2022. PDCs were established from pleural effusion, pericardial effusion, ascites, and tumor tissue samples. Tumor fluid samples were transferred to conical tubes and centrifuged at 120× *g* for 10 min. Each cell pellet was resuspended in the Roswell Park Memorial Institute (RPMI 1640, Corning, Manassas, VA, USA) medium and carefully layered onto a lymphocyte separation medium (#091692249; MP Biomedicals, Solon, OH, USA). After the suspension was centrifuged at 400× *g* for 30 min at room temperature, the tumor cells were trapped in the interphase layer between the RPMI 1640 medium and the lymphocyte separation medium; these cells were harvested and washed with the RPMI 1640 medium. The tumor tissue samples were finely minced and washed with the RPMI 1640 medium. Cells released into the washing fluid or from the tissue were cultured in the AR-5 medium (5% fetal bovine serum (Corning, Manassas, VA, USA), 1 × GlutaMAX (Thermo Fisher Scientific, Waltham, MA, USA), 1× ITS (insulin–transferrin–selenium, Thermo Fisher Scientific), 1% penicillin/streptomycin, 50 nM of hydrocortisone, 1 mM of sodium pyruvate, and 1 ng/mL of epidermal growth factor in RPMI 1640 medium) at 37 °C under 5% CO_2_. The medium was carefully exchanged every 2–3 days until cells stabilized on the bottom of the flask.

### 2.2. Drug Sensitivity Screening Using PDCs

For each treatment, stabilized PDCs were seeded in 384-well plates (1000 cells/well in 20 μL aliquots) in quadruplicate. Simultaneous tests were conducted for 64 chemotherapeutic agents that are commonly used to treat lung cancer or have shown promising efficacy in preclinical studies (Appendix A). After overnight incubation, the cells were treated with one drug at a 5-fold serial dilution for a total of six doses (50 μM to 16 nM). Cell viability was measured after 72 h of treatment using the CellTiter-Glo Luminescent cell viability assay kit (Promega, Madison, WI, USA) and an Infinite 200 Pro system (TECAN, Mannedorf, Switzerland). Each screening plate contained a dimethyl-sulfoxide-only vehicle control to calculate the relative cell viability and normalize the data. Dose–response curve fitting and area under the curve (AUC) values were assessed using GraphPad Prism 5.3 (GraphPad Software Inc., San Diego, CA, USA). The relative sensitivity of each drug was determined by calculating standardized AUCs based on means and standard deviations, using Excel (Microsoft Office 2013 Excel).

### 2.3. Cell Line and Reagents

Human lung cancer cell lines were used as a reference for drug sensitivity. PC9 cells, as lung cancer cells with an *EGFR* exon 19 deletion mutation, were purchased from RIKEN BioResource Center Cell Bank (Ibaraki, Japan). SNU-2292 cells, as lung cancer cells with an *ALK* fusion, were obtained from the Korea Cell Bank (Seoul, Republic of Korea). H1299 cells and A549 cells, as lung cancer cells harboring wild-type *EGFR* and *ALK*, respectively, were purchased from the American Type Culture Collection (Manassas, VA, USA) and the Korea Cell Bank (Seoul, Republic of Korea). All cell lines were maintained in RPMI 1640 medium with 10% fetal bovine serum. Cell-line authentication was performed annually at the Genomic Core Facility of the National Cancer Center. Compounds included in the screening-compound library were newly prepared each month and tested for the preservation of chemical activities using NSCLC cancer cell lines (A549, PC9, and H1299). All library compounds were purchased from Selleckchem (Houston, TX, USA).

### 2.4. Clinical Data Collection

Medical records and radiographic images were reviewed; a predesigned data collection format was used to collect data regarding the clinicopathological characteristics of the tumor, tumor response, and patient survival outcomes. The tumor response was generally assessed during chemotherapy at intervals of 6–12 weeks, via computed tomography and other imaging techniques, in accordance with guidelines established by the Response Evaluation Criteria in Solid Tumors (RECIST) Committee 1.1 [18].

### 2.5. Targeted Next-Generation Sequencing (NGS) of PDCs

Genomic DNA was extracted from cultured cells using the AllPrep DNA/RNA mini kit (Qiagen, Hilden, Germany) and processed on a QIAcube automatic instrument (Qiagen). Fragmented genomic DNA was hybridized with RNA probes from the SureSelect XT custom panel kit library and sequenced on an Illumina NovaSeq 6000 platform (Illumina, San Diego, CA, USA). Images were analyzed using the NovaSeq6000 control software, v1.3.1, and output base calling data were de-multiplexed with bcl2fastq v2.20.0.422 by generating fastQC files.

### 2.6. Statistical Analysis

The Pearson χ^2^ test and Fisher’s exact test were used to determine relationships between categorical variables, as appropriate. Relationships between categorical variables and continuous variables were assessed using the Mann–Whitney U test or a one-way analysis of variance (ANOVA). Progression-free survival (PFS) was calculated from treatment initiation to the first documentation of disease progression, death, or the last follow-up visit. Overall survival (OS) was calculated from treatment initiation to death or the last follow-up visit. Survival differences were estimated using the Kaplan–Meier method; survival differences were compared between groups using the log-rank test. Cox proportional hazards models were used to calculate hazard ratios for survival. Two-sided *p*-values < 0.05 were assumed to indicate statistical significance.

## 3. Results

### 3.1. Establishment of PDC Drug Screening Platform

PDCs were established using malignant effusion and/or tissue samples from patients with advanced lung cancer, with a success rate of 81.5% (397/487, Figure 1A, Appendix A). The cultures were used to assess sensitivity to 64 anti-cancer drugs (Appendix A) within less than one cell culture passage. The median time from sample collection to drug response report was 15.5 days (range, 4–64 days). Most patients had adenocarcinoma (n = 320 [81%]) previously treated with chemotherapy or targeting agents (n = 279 [70%]). The PDCs were derived from patients who received targeted therapy (n = 182, 45.8%) or non-targeted chemotherapy (n = 215, 54.2%) (Figure 1B). There were 43 (23.6%) treatment-naïve patients in the targeted therapy group and 75 (34.9%) treatment-naïve patients in the non-targeted chemotherapy group.

### 3.2. Association between PDC Drug Sensitivity and Clinical Tumor Response

A comparison of the PDC drug sensitivity with the clinical tumor response (Figure 2A) was performed using 279 PDCs for which both in vitro drug response data and subsequent chemotherapy outcome and follow-up data were available (Figure 2A). These cases were divided into two groups: targeted treatment and non-targeted treatment. The targeted treatment group included 139 PDCs from treatment-naïve patients whose tumors contained *EGFR* (n = 103), *HER2* (n = 3), or *BRAF V600E* (n = 1) mutations; *ALK* fusion (n = 17); or *MET* amplification (n = 16). Sensitivity to the targeting agent was determined by calculating the standardized AUC value of the drug. In this group, the standardized AUC values of the PDCs were significantly associated with the clinical tumor response (one-way ANOVA, *p* = 0.002). AUC values were significantly higher in the progressive disease (PD) subgroup than in the stable disease (SD) or partial response (PR) subgroups (median standardized AUC values: 0.23 [95% confidence interval (CI), −0.16 to 0.55] in PD, −0.26 [95% CI, −0.44 to 0.14] in SD, and −0.31 [−0.87 to −0.29] in PR). The AUC values of PDCs in the cytotoxic chemotherapy group were not associated with the clinical tumor response (one-way ANOVA, *p* = 0.973).

Attempts to obtain long-term PDCs (>10 passages) to establish permanent cell lines and thus evaluate changes in drug sensitivity over time were successful for only 49 cultures (12.3%). A response comparable with the patient’s tumor response to targeted therapy was obtained in 18 of these PDCs. In the other PDCs, the drug sensitivity changed in a variable manner after long-term passage. In four long-term PDCs established from patients who developed acquired resistance to the drugs (PD subgroup), the AUC values of the targeted drugs were lower than in the initial short-term PDCs (Figure 2B). For example, in a 77-year-old female patient (case NCCLu-261) with stage IV lung adenocarcinoma displaying *EGFR* 19 deletion and acquired resistance to erlotinib after an initial response, the short-term PDC was insensitive to gefitinib but the long-term PDC was sensitive (standardized AUCs = 1.21 and −3.82, respectively; Figure 2C). This finding suggests that the true clinical outcome will be better reflected by the short-term PDC rather than the long-term PDC.

### 3.3. PDC Drug Sensitivity Predicts Clinical Outcome in Treatment-Naive EGFR- or ALK-Positive NSCLC

The predictive role of PDC drug sensitivity in the targeting agent group was further explored in treatment-naive patients whose tumors harbored *EGFR* mutations (n = 49) or *ALK* fusion (n = 10) (Appendix A). The median patient age was 64 years (range, 37–84 years), and 61.0% of the patients were women (Appendix A). Adenocarcinoma was the predominant histologic type (94.9%), and most PDCs were derived from effusions (93.2%). PR, SD, and PD were achieved in 39 (66%), 10 (17%), and 10 (17%) patients, respectively. The median PFS was 8.6 months (95% CI, 7.0–10.0 months), and the median OS was 21.5 months (95% CI, 17.0–25.9 months). A comparison of the PDC drug sensitivity and clinical tumor response showed that the PDC AUC values for EGFR- or ALK-TKIs were significantly associated with the clinical tumor response (median standardized AUC values: −0.78 in PR, −0.29 in SD, and 0.30 in PD; *p* < 0.001; Figure 3A). The waterfall plot similarly showed a strong trend between PDC AUC values and the clinical tumor response to EGFR- or ALK-TKIs (Figure 3B).

The patients were divided into two subgroups according to the baseline responses of their PDCs to the corresponding TKIs. PDC responder patients were defined based on a standardized AUC value of ≤50% in their PDC; PDC non-responders were defined based on a standardized AUC value of > 50% in their PDC (Appendix A). The PDC non-responder group (n = 19) had a significantly lower tumor response rate compared with the PDC responder group (n = 40) (25.0% vs. 77.5%, *p* = 0.011; Figure 3C). The median PFS was significantly shorter among PDC non-responders than among PDC responders (3.4 months [95% CI, 2.8–4.1 months] vs. 11.8 months [95% CI, 6.5–17.0 months]; *p* < 0.001) (Figure 3D). A trend toward a shorter OS was also identified in the PDC non-responder group compared with the PDC responder group (11.9 months [95% CI, 2.4–26.0 months] vs. 27.2 months [95% CI, 17.7–36.6 months]; *p* = 0.060) (Figure 3E). These trends were observed in both the subgroup positive for *EGFR* mutations and in the subgroup positive for *ALK* fusion (Appendix A). The predictive role of the PDC response to the targeting agent was evaluated with respect to well-established predictive factors, including age, sex, Eastern Cooperative Oncology Group (ECOG) performance status, smoking history, histology, operation history, brain metastasis status, and *TP53* mutation status (Table 1 and Appendix A). In the multivariate analysis of PFS, the baseline PDC response was independently associated with a better PFS in untreated patients with *EGFR/ALK*-positive lung cancer (hazard ratio: 0.20, 95% CI: 0.07–0.0.60, *p* = 0.004). Genetic alterations in PDCs were also analyzed to identify potential resistance mechanisms, but there were no significant differences in genetic profiles between PDC responders and non-responders (Appendix A).

### 3.4. Causes of a Poor Response to EGFR- or ALK-TKIs

Mechanisms underlying poor clinical responses to EGFR- or ALK-TKIs were examined by analyzing the clinical, pathological, and genomic characteristics of the 10 PDC non-responders (Appendix A). Three patients had SD (30%) and seven patients had PD (70%) as the best response to first-line TKIs; the median PFS was 1.7 (95% CI, 0.1–3.7) months and the median OS was 6.0 (95% CI, 1.4–10.6) months. A comprehensive analysis revealed several potential causes of resistance to EGFR- or ALK-TKIs, described in the following sections.

#### 3.4.1. Concurrent Resistant Gene Alterations

A 63-year-old male patient with stage IV lung adenocarcinoma displaying *EGFR* 19 deletion (case NCCLu-263) received erlotinib as the first-line treatment (Figure 4A–D). Targeted NGS with plasma circulating tumor DNA (ctDNA) detected *PIK3CA* M1004I and the *EGFR* 19 deletion. The baseline PDC was established on day 9 after sampling (Figure 4A). Targeted NGS of the baseline PDC also identified *PIK3CA* M1004I and the *EGFR* 19 deletion. Baseline PDC drug sensitivity screening showed insensitivity to gefitinib (standardized AUC value, 0.23; Figure 4B). The patient’s best response to first-line erlotinib was SD, but his disease progressed after 3.1 months (Figure 4D). Treatment consisting of an EGFR-TKI together with a PIK3CA inhibitor was investigated using the PDC sample, which showed synergism of the two drugs in terms of inhibiting the growth of cells harboring the identified *EGFR* and *PIK3CA* mutations (Figure 4C).

#### 3.4.2. Intra-Tumoral Heterogeneity

A 60-year-old male patient (PDC case-131) with stage IV lung adenocarcinoma displaying *ALK* fusion received crizotinib (Figure 4E–H). Multiple genetic tests of a biopsy sample of the left axillary metastatic lymph nodes were performed at the time of diagnosis. A companion fluorescence in situ hybridization assay of the biopsy samples revealed *ALK* fusion (Figure 4H). However, after 4 weeks, targeted NGS of the same tumor tissue showed a *BRAF* V600E mutation but no evidence of an ALK fusion. The results of the patient’s baseline PDC NGS and the PDC drug sensitivity test are shown in Figure 4E. The baseline PDC NGS results were consistent with the tumor NGS data; they also showed a *BRAF* V600E mutation without an *ALK* fusion. The PDC drug screening test revealed insensitivity to crizotinib (standardized AUC value, 0.07; Figure 4F). The patient began crizotinib treatment for *ALK*-positive lung cancer, but metastasis increased in the left axillary and right mediastinal metastatic lymph nodes; the patient’s disease progressed after 4 weeks of treatment (Figure 4H). Because the tumor findings, PDC NGS analysis, and PDC drug sensitivity test suggested both *ALK*-positive cells and *BRAF*-mutated malignant cells, the combined effect of an ALK-TKI and a BRAF inhibitor was tested on the PDC. The results showed synergistic action of the two drugs (Figure 4G). After treatment with dabrafenib plus trametinib, SD was identified in the left axillary and mediastinal metastatic lymph nodes. However, the patient discontinued treatment after 6 weeks because of severe pneumonitis.

#### 3.4.3. Inter-Lesional Tumor Heterogeneity

A 37-year-old female patient (case NCCLu-089) was diagnosed with stage IV lung adenocarcinoma displaying *ALK* fusion (Figure 4I–K). The *ALK* fusion was detected in her mediastinal lymph nodes by immunohistochemistry and targeted NGS. However, in the baseline PDC (derived from the malignant pleural effusion), the *ALK* fusion was not detected; the drug screening test showed insensitivity to ceritinib (standardized AUC value, 0.29; Figure 4I,J). After 8 weeks of alectinib treatment, *ALK*-fusion-positive cells in the patient’s mediastinal lymph nodes decreased but *ALK* fusion-negative malignant cells increased in the pleural effusion (Figure 4K). Subsequent treatment consisted of pemetrexed and cisplatin, but the patient’s disease did not respond to either drug.

#### 3.4.4. Histology

A 58-year-old female patient (case NCCLu-237) was diagnosed with stage IV large-cell neuroendocrine carcinoma displaying *ALK* fusion. An initial genetic analysis of the liver metastasis showed *ALK* fusion, confirmed by immunohistochemistry and targeted NGS. However, the baseline PDC drug screening test showed insensitivity to alectinib (standardized AUC value, 0.08). Based on the companion immunohistochemistry results, the patient began alectinib treatment but her disease immediately progressed. After alectinib failure, the *ALK* fusion statuses of the primary tumor and pleural effusion were re-evaluated, but the genetic alteration remained detectable in both specimens.

### 3.5. PDC-Based Prediction of an Unexpected Response to EGFR-Targeted Drugs

Among the 397 PDCs subjected to drug screening tests, 47 had a high sensitivity to EGFR-TKIs (rank standardized AUC value, ≤25%; Figure 5A and Appendix A). Common drug-sensitive *EGFR* mutations, including exon 19 deletion and exon 21 L858R, were detected in 30 PDCs; wild-type *EGFR* was detected in 17 PDCs and an uncommon *EGFR* mutation (exon 20 insertion) was detected in one PDC. Among the 30 EGFR-TKI-sensitive PDCs with sensitive *EGFR* mutations, eight were established before EGFR-TKI exposure and 22 were established after EGFR-TKI treatment. All eight EGFR-TKI-naïve patients had a PR to the EGFR-TKIs (eight of eight, 100%). Among the 22 patients who previously received EGFR-TKIs, 11 were re-treated with EGFR-TKIs. Two patients received post-third-generation EGFR-TKIs, whereas nine patients received post-first-generation EGFR-TKIs. None of the tumor cells from these patients showed evidence of a T790M mutation. Five (45%) patients had PRs after the resumption of EGFR-TKI treatment; among 13 patients re-treated with EGFR-TKIs despite PDC insensitivity to the same drugs, the PR was 15% (2 of 13) (Figure 5B). After EGFR-TKI rechallenge, the disease control rates in the EGFR-TKI-sensitive and EGFR-TKI-insensitive PDC groups were 63% and 38%, respectively. A 58-year-old female patient (case NCCLu-045) with stage IV lung adenocarcinoma harboring *EGFR* L858R received two EGFR-TKIs (erlotinib and olmutinib) and multiple rounds of chemotherapy. However, her PDC was highly sensitive to EGFR-TKIs (Figure 5C–E). PDC drug screening of the malignant effusion collected after docetaxel treatment showed high sensitivity to gefitinib, afatinib, and osimertinib. These results were comparable with the results of in vitro tests using PC9 cells, a lung cancer cell line with an *EGFR* 19 deletion sensitive to EGFR-TKIs (standardized AUCs for gefitinib, afatinib, and osimertinib: −4.39, −3.15, and −3.21, respectively). Based on the PDC result, the patient was re-treated with erlotinib, which led to a PR, 50% tumor shrinkage, and 11 months of PFS.

Among the 17 EGFR-TKI-sensitive PDCs with wild-type *EGFR* or uncommon *EGFR* mutations, four of the respective patients received EGFR-TKIs, resulting in PR in two of them (Figure 5A). One patient (case NCCLu-027) was a never-smoking 84-year-old woman with stage IV lung adenocarcinoma lacking driver oncogenes such as *EGFR* (Figure 5F–H). Genetic tests for *EGFR* mutations, including a commercial kit (Cobas^®^ mutation assay) and targeted NGS, were performed six times; the results were consistently negative with respect to an *EGFR* mutation (Appendix A). However, the patient’s PDC was highly sensitive to all three EGFR-TKIs, similar to the results of in vitro tests using PC9 cells (standardized AUCs for gefitinib, afatinib, and osimertinib: −3.05, −2.18, and −2.22, respectively; Figure 5G). Therefore, the patient was treated with erlotinib, which resulted in a PR for 22 months (Figure 5H). Ultra-deep sequencing was thus performed to detect a drug-sensitive *EGFR* mutation in the patient’s tumor cells (Appendix A). A droplet digital polymerase chain reaction analysis of pretreatment plasma ctDNA revealed an *EGFR* L858R mutation with a 1.4% fractional abundance.

The other patient (case NCCLu-157) was a 61-year-old man diagnosed with stage IV lung adenocarcinoma harboring *EGFR* exon 20 A763_Y764insFQEA (Figure 5I–K). At the time of diagnosis, *EGFR* exon 20 insertion-specific targeted therapy was not available. Accordingly, the patient received gemcitabine and carboplatin as the first-line treatment, which induced a PR (Figure 5I); however, his disease progressed after 6 months. A PDC was established using the malignant pleural effusion; a subsequent drug sensitivity test showed high sensitivity to gefitinib, afatinib, and osimertinib (standardized AUC: −1.57, −2.28, and −1.42, respectively; Figure 5J). Based on the PDC response, the patient received erlotinib, resulting in a PR for 8 months (Figure 5K). After disease progression and a switch to erlotinib, tissue NGS was performed, which revealed *EGFR* T790M present in *cis* with exon 20 A763_Y764insFQEA. Therefore, the patient began osimertinib treatment, which induced a good response in the resistant tumor (Appendix A). Further drug sensitivity testing to common EGFR-TKIs using PDCs with variable *EGFR* exon 20 insertions (Figure 5L) showed a variable response according to the *EGFR* exon 20 insertion type. Only the A763_Y764insFQEA mutation was sensitive to common EGFR-TKIs. Among six patients with PDCs harboring *EGFR* 20 insertions insensitive to common EGFR-TKIs, four were treated with erlotinib (n = 3) or afatinib (n = 1) but none had a response.

## 4. Discussion

Although PDC models have been extensively studied as a tool for cancer drug screening, long-term PDCs lose the phenotypic characteristics of the primary tumor, hindering the prediction of clinical outcomes. In this study, 397 lung cancer PDCs were established through short-term culturing, and their sensitivities to different classes of anti-cancer drugs were tested. Additionally, the pharmacogenomic data of the PDCs were matched with the respective patient’s clinical data, allowing for an evaluation of the clinical utility of the PDC model in the management of lung cancer. Our study demonstrated that the drug sensitivity of PDCs to targeting agents was closely associated with the patient’s tumor response; the drug sensitivity of the pretreatment PDC was strongly predictive of the clinical response to EGFR- or ALK-TKIs in patients with *EGFR*- or *ALK*-positive lung cancer. Patients with pretreatment PDC insensitivity to the matched TKIs had a shorter PFS compared with patients whose pretreatment PDC displayed high sensitivity to those drugs. Analyses of genomic and functional data from PDCs may reveal contributing factors that distinguish non-responders from responders to EGFR- or ALK-TKIs in the absence of conclusive genomic data.

Although validated genetic tests are used to identify patients whose tumors are likely to respond to EGFR- or ALK-targeted drugs, ~20% of *EGFR*- and *ALK*-positive NSCLC patients will be non-responders [19,20,21]. In the present study, 31.5% of patients with untreated *EGFR*- or *ALK*-positive lung cancer were non-responsive to matched EGFR- or ALK-targeted therapy; accordingly, they had poor PFS (2.8 months, 95% CI, 1.5–4.1 months), indicative of persistent deficiencies in current genomic-data-guided targeted therapy. Our study demonstrated various issues underlying the clinically unexpected absence of a response to targeting agents and the potential utility of the PDC drug screening platform in predicting clinical outcomes. In case NCCLu-263, the Cobas^®^ EGFR mutation test identified an *EGFR* 19 deletion mutation in the tumor, but the patient did not display dramatic improvement after receipt of erlotinib. Indeed, the patient’s baseline PDC drug screening result indicated insensitivity to EGFR-TKIs. Targeted NGS analyses of the tumor and PDC both identified a *PIK3CA* M1004I mutation and an *EGFR* 19 deletion, which together may have caused the resistance to erlotinib. Finally, the same PDC showed synergistic effects of an EGFR-TKI in combination with a PIK3CA inhibitor. Among lung cancers with *EGFR*-sensitive mutations, 3.5% harbor the *PIK3CA* mutation [22]; however, its influence on EGFR-TKI efficacy in *EGFR*-mutated lung cancer is controversial [9]. Some studies showed that patients with tumors harboring a concomitant *PIK3CA* mutation have a lower RR to EGFR-TKIs, as well as shorter PFS and OS durations [23]; other studies revealed longer PFS and OS [24]. Jin et al. reported a domain-specific effect of the *PIK3CA* mutation on PFS in patients treated with EGFR-TKIs; this finding indicated that p85-binding domain mutations (including R108H) are associated with a longer PFS, whereas kinase (Y1021H), helical (E542K), and C2 (N345K) domain mutations are associated with a shorter PFS [25]. In the present study, the tumor in case NCCLu-263 had a *PIK3CA* kinase domain (Y1021H) mutation and therefore did not respond to erlotinib.

In lung cancer, the major histologic type associated with an *EGFR* or *ALK* mutation is adenocarcinoma [26]. In rare cases, these driver oncogenic mutations are identified in squamous cell carcinoma or high-grade neuroendocrine cell carcinoma [27,28]. In a recent report by Lam et al., *EGFR* mutation and *ALK* fusion were detected in 2.7% and 0.5% of squamous cell lung carcinomas, respectively [29]. In the present study, the tumors in cases NCCLu-237 and NCCLu-253 consisted of an *ALK*-positive large-cell neuroendocrine carcinoma and an *EGFR*-mutant squamous cell lung carcinoma, respectively. In these lung cancers with atypical histology, the benefit of targeted therapy is limited; this limited benefit was anticipated by the insensitivity of the PDC to the corresponding targeted drugs. Taken together, these results suggest that pretreatment PDC drug screening can provide more precise and individualized information. Patients with *EGFR-* or *ALK*-positive lung cancer whose pretreatment PDC indicates insensitivity to the matched targeted drugs will need more careful monitoring for earlier disease progression; this will enable planning for a possible alternative treatment, such as combination therapy or a modified regimen.

Another clinical application of the PDC drug screening platform is the selection of the appropriate anti-cancer drugs in lung cancer patients. In case NCCLu-131, intra-tumoral heterogeneity was caused by two actionable driver mutations—*ALK* fusion and a *BRAF* V600E mutation—but it was unclear whether they should be targeted sequentially or simultaneously. The patient’s baseline PDC did not respond to either ALK inhibitors or BRAF inhibitors; instead, it suggested that the combination of an ALK-TKI and a BRAF inhibitor would be the most effective. In case NCCLu-027, the patient’s tumor was diagnosed as wild-type *EGFR* but showed a strong response to an EGFR inhibitor. Neither conventional genetic tests (plasma and tissue Cobas^®^) nor targeted NGS detected an *EGFR* mutation. However, the responsiveness of the patient’s PDC to EGFR inhibitors correctly predicted a benefit from EGFR-targeted therapy. Similarly, in case NCCLu-045, the patient’s tumor successfully responded to re-treatment with an EGFR-TKI. In *EGFR*-mutant lung cancers, re-treatment with an EGFR-TKI is regarded as a treatment option, although the RR (5–25%) is much lower than the RR in treatment-naïve tumors (70–80%) [30,31,32]. Predictive markers of relapsed tumors that will exhibit a durable response to EGFR-TKI re-treatment are lacking. The present study demonstrated that the RR was significantly higher in patients with an EGFR-TKI-sensitive PDC than in patients with an EGFR-TKI-insensitive PDC during re-treatment with EGFR-TKIs. Therefore, the PDC drug screening test may serve as a predictive tool to identify patients with previously treated EGFR-mutant NSCLC who will benefit from EGFR-TKI re-treatment.

The *EGFR* A763_Y764insFQEA mutation involves the insertion of 5% of *EGFR* exon 20 [33]. There is emerging evidence that the presence of this mutation indicates tumor sensitivity to classic EGFR-targeting inhibitors, in contrast to other *EGFR* exon 20 insertion subtypes [21,33]. Yasuda et al. determined that the difference reflects similar structural and kinetic features in this mutation and EGFR-TKI-sensitive mutations (e.g., exon 19 deletion and L858R mutations) [34]. In the present study, the results of case NCCLu-157, in which the patient’s tumor cells harbored the *EGFR* A763_Y764ins mutation, were consistent with previous reports. The pretreatment PDC harboring *EGFR* A763_Y764ins was highly sensitive to gefitinib, afatinib, and osimertinib; PDCs with other *EGFR* exon 20 insertion types did not display these sensitivities. The patient’s tumor responded well to erlotinib but developed resistance after the acquisition of the T790M mutation, the most common type of resistance to first- or second-generation EGFR-TKIs. These results suggest that the pharmacogenomic features of the PDC closely recapitulate the biological behaviors of tumors with rare oncogenic mutations. Therefore, the PDC drug screening test may provide highly informative data that can facilitate the treatment of rare tumors with minimal relevant published data.

## 5. Conclusions

The PDC drug screening platform is a rapid and feasible tool for the treatment of advanced lung cancer. Our study showed that it can be successfully used to guide targeted therapy for lung cancer patients with a poor prognosis; it may overcome the limitations of genomic assay-based targeted therapy. An approach integrating clinical–genomic data with functional–genomic data from PDCs will likely result in more effective and tailored anti-cancer treatments for lung cancer patients.

## Figures and Tables

**Figure 1 cancers-16-00778-f001:**
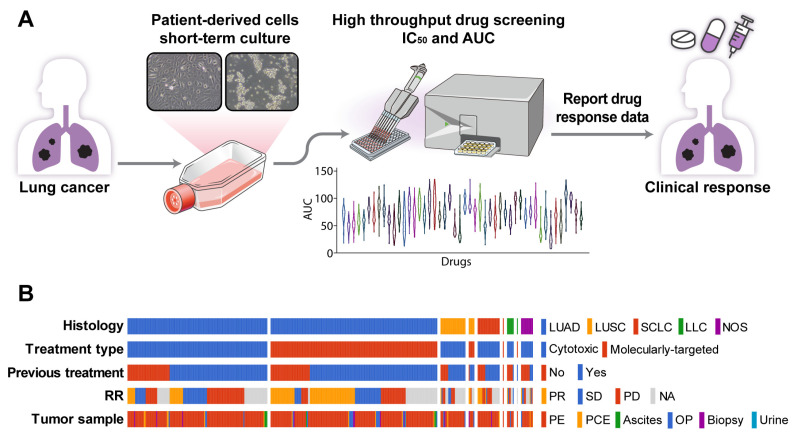
PDC drug screening platform. (**A**) Overview of the PDC drug screening platform. (**B**) Summary plot of a comprehensive analysis of PDC drug responses and clinical outcomes based on 397 PDCs. Abbreviations: LUAD, lung adenocarcinoma; LUSC, lung squamous cell carcinoma; SCLC, small-cell lung carcinoma; LCC, large-cell carcinoma; NOS, other histologic-type tumor; RR, response rate; PR, partial response; SD, stable disease; PD, progression disease; NA, not analyzed; PE, pleural effusion; PCE, pericardial effusion; OP, operation tissue.

**Figure 2 cancers-16-00778-f002:**
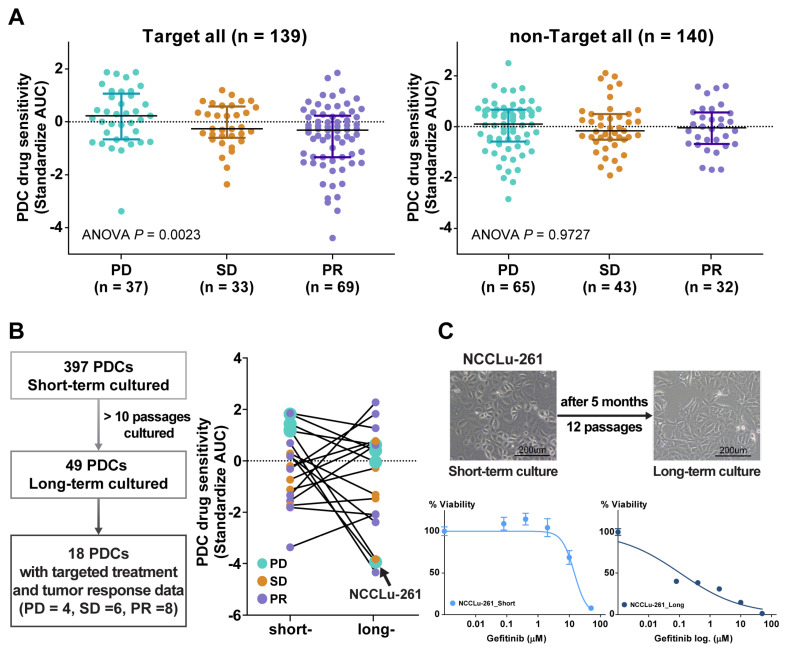
(**A**) Scatter plot of the standardized AUC values of drugs according to tumor response from the PDCs of 139 patients who received targeted therapy and 140 patients who received non-targeted therapy. (**B**) Scatter plot of the standardized AUC values of short-term and long-term PDCs. (**C**) Bright-field microscopy images of PDC NCCLu-261 after 7 days (short-term) and 5 months (long-term). Scale bar, 200 μm. Viability curve of gefitinib-treated cells from PDC NCCLu-261 in short-term and long-term culture.

**Figure 3 cancers-16-00778-f003:**
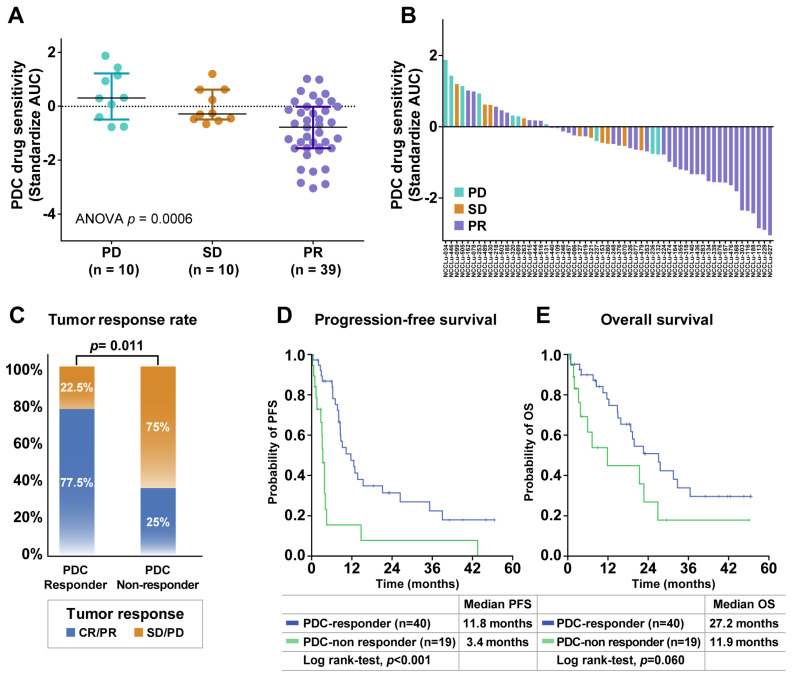
Associations of PDC drug response with clinical outcomes after targeted therapy. (**A**) Scatter plot of standardized AUC values from PDCs testing *EGFR*- or *ALK*-targeted drugs according to the tumor response in 59 chemo-naive *EGFR*- or *ALK*-positive NSCLC patients. Horizontal lines represent the 0.25, 0.50 (black), and 0.75 quantiles. The *p* value was calculated by ANOVA. (**B**) Waterfall plot of standardized AUC values from the PDCs as shown for each drug. (**C**) Tumor response rates between PDC responders and PDC non-responders. The *p* value was calculated by Pearson χ^2^ test. (**D**,**E**) Kaplan–Meier curves for progression-free survival (PFS) and overall survival (OS) of PDC responders and PDC non-responders. Abbreviations: PD, progressive disease; SD, stable disease; PR, partial response.

**Figure 4 cancers-16-00778-f004:**
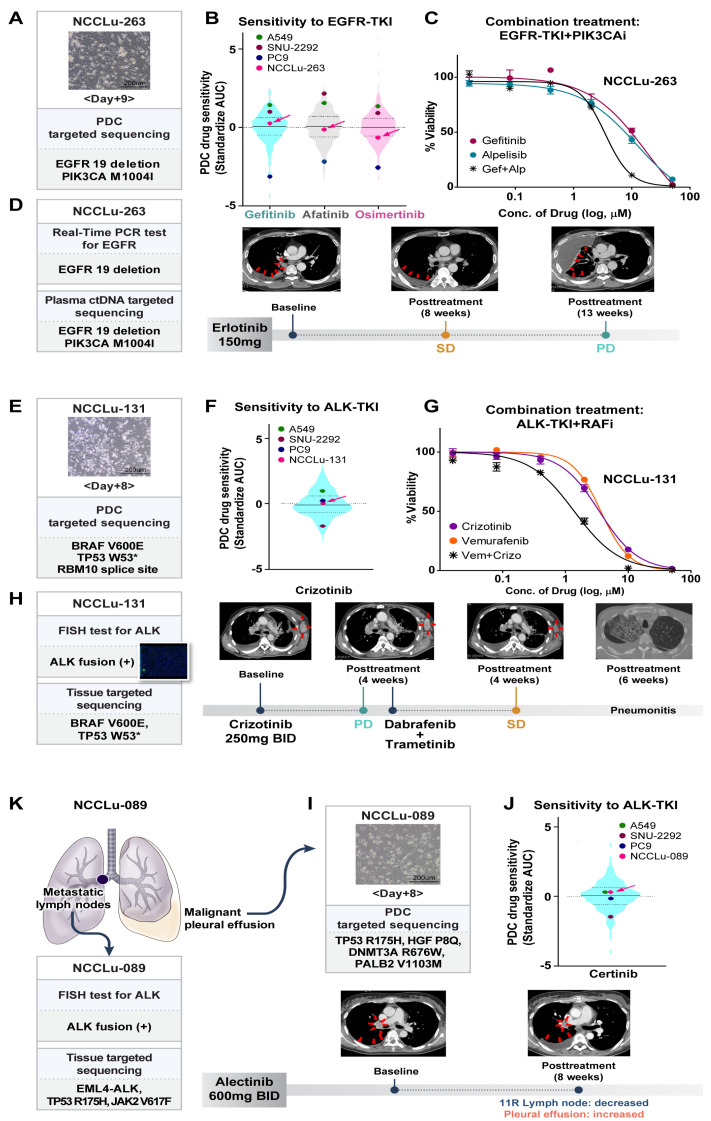
PDC predictions of unexpected non-responders to targeted therapy. (**A**–**D**) Case NCCLu-263. (**A**) Bright-field microscopy images of the PDC after 9 days of culture. Scale bar, 200 μm. (**B**) Violin plots of the standardized AUC values of EGFR-TKIs based on 396 PDCs. Each dot indicates the AUC value of PDC NCCLu-263 or the control cell lines (positive control, PC9 with *EGFR* 19 deletion; negative control, A549 and SNU-2292 with wild-type *EGFR*). Horizontal lines represent the 0.25, 0.50, and 0.75 quantiles. (**C**) Viability curves of cells from PDC NCCLu-263 treated with gefitinib, alpelisib, or both. (**D**) Chest computed tomography images of the patient before and after treatment. Red arrows indicate the tumor. (**E**–**H**) Case NCCLu-131. (**E**) Bright-field microscopy images of the PDC after 8 days of culture. Scale bar, 200 μm. (**F**) Violin plots of the standardized AUC values of crizotinib based on 386 PDCs. Each dot indicates the AUC values of PDC NCCLu-131 or the control cell lines (positive control, SNU-2292 with *ALK* fusion; negative control, A549 and PC9 with wild-type *ALK*). Horizontal lines represent the 0.25, 0.50, and 0.75 quantiles. (**G**) Viability curves of cells from NCCLu-131 PDC treated with crizotinib, vemurafenib, or both. (**H**) Chest computed tomography images of the patient before and after treatment. Red arrows indicate the tumor. (**I**–**K**) Case NCCLu-089. (**I**) Bright-field microscopy images of the PDC after 8 days of culture. Scale bar, 200 μm. (**J**) Violin plots of the standardized AUC values of ceritinib based on 381 PDCs. Each dot indicates the AUC values of PDC NCCLu-089 or control cell lines (positive control, SNU-2292 with *ALK* fusion; negative control, A549 and PC9 with wild-type *ALK*). Horizontal lines represent the 0.25, 0.50, and 0.75 quantiles. (**K**) Chest computed tomography of the patient before and after treatment. Red arrows indicate the tumor. Abbreviations: EGFR-TKI, EGFR tyrosine kinase inhibitor; PIK3CAi, PIK3CA inhibitor; RAFi, RAF inhibitor; SD, stable disease; PD, progressive disease.

**Figure 5 cancers-16-00778-f005:**
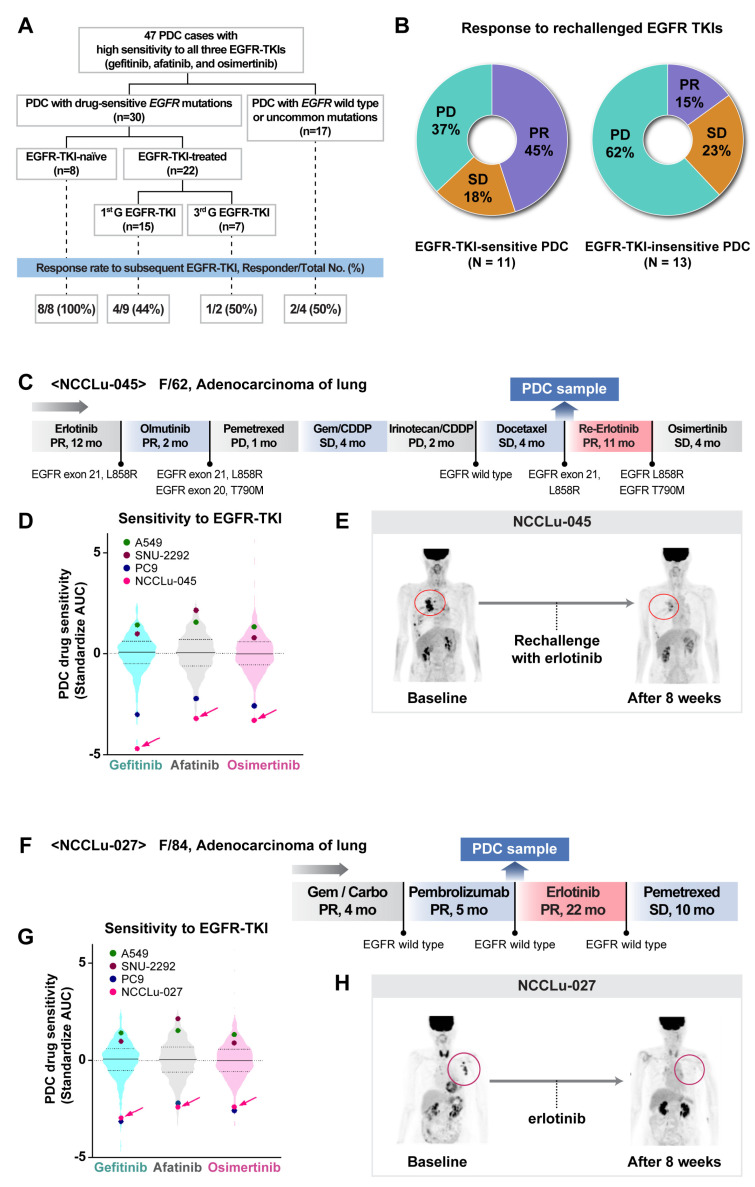
PDC prediction of unexpected responders to EGFR-targeted therapy. (**A**) Consort diagram of 47 PDCs showing high sensitivity to all three EGFR-TKIs. (**B**) Pie plots of the percentages of responders among patients rechallenged with EGFR-TKIs after resistance to previous EGFR-TKI treatment for *EGFR*-mutant NSCLC. (**C**–**E**) Case NCCLu-045. (**C**) Clinical course. (**D**) Violin plots of the standardized AUC values of gefitinib, afatinib, and osimertinib based on 396, 391, and 229 PDCs, respectively. Each dot indicates the AUC values of PDC NCCLu-045 or control cell lines (positive control, PC9 with *EGFR* 19 deletion; negative control, A549 and SNU-2292 with wild-type *EGFR*). Horizontal lines represent the 0.25, 0.50, and 0.75 quantiles. (**E**) Positron emission tomography images of the patient before and after treatment. Red circles indicate the tumor. (**F**–**H**) Case NCCLu-027. (**F**) Clinical course. (**G**) Violin plots of the standardized AUC values of gefitinib, afatinib, and osimertinib based on 396, 391, and 229 PDCs, respectively. Each dot indicates the AUC values of PDC NCCLu-027 or the control cell lines (positive control, PC9 with *EGFR* 19 deletion; negative control, A549 and SNU-2292 with wild-type *EGFR*). Horizontal lines represent the 0.25, 0.50, and 0.75 quantiles. (**H**) Positron emission tomography images of the patient before and after treatment. Red circles indicate the tumor. (**I**–**L**) Case NCCLu-157. (**I**) Clinical course. (**J**) Violin plots of the standardized AUC values of gefitinib, afatinib, and osimertinib based on 396, 391, and 229 PDCs, respectively. Each dot indicates the AUC values of PDC NCCLu-157 or control cell lines (positive control, PC9 with *EGFR* 19 deletion; negative control, A549 and SNU-2292 with wild-type *EGFR*). Horizontal lines represent the 0.25, 0.50, and 0.75 quantiles. (**K**) Chest computed tomography and positron emission tomography images of the patient before and after treatment. Red circles indicate the tumor. (**L**) Waterfall plot of the standardized AUC value of each classic EGFR-TKI as determined in seven PDCs in which the *EGFR* exon 20 insertion was identified. Lower panel: actual tumor response to classic EGFR-TKIs in each patient. Abbreviations: EGFR-TKI, EGFR tyrosine kinase inhibitor; 1st G EGFR-TKI, first-generation EGFR-TKI; 3rd G EGFR-TKI, third-generation EGFR-TKI; PR, partial response; SD, stable disease; PD, progressive disease. Associations of PDC drug response with clinical outcomes after targeted therapy.

**Table 1 cancers-16-00778-t001:** Multivariate analysis of progression-free survival for EGFR- or ALK-tyrosine.

Characteristics		No.	Univariate Analysis		Multivariate Analysis	
			HR (95% CI)	*p* ^†^	HR (95% CI)	*p* ^†^
Age	≥65 vs. <65 years	28 vs. 31	0.81 (0.43–1.52)	0.507		
Gender	Male vs. Female	23 vs. 36	1.70 (0.87–3.32)	0.119		
ECOG PS	0–1 vs. 2–3	41 vs. 18	0.41 (0.21–0.81)	0.010	0.56 (0.28–1.13)	0.106
Smoking	Never vs. Ever	30 vs. 19	0.73 (0.38–1.38)	0.327		
Histology	ADC vs. non-ADC	56 vs. 3	0.20 (0.06–0.70)	0.012	0.43 (0.11–1.69)	0.227
Operation	No vs. Yes	42 vs. 12	2.71 (1.25–5.91)	0.012		
BM metastasis	No vs. Yes	41 vs. 18	0.51 (0.26–1.02)	0.056	0.87 (0.40–1.88)	0.722
*TP53*	WT vs. MT	14 vs. 11	0.71 (0.29–1.72)	0.451		
PDC response	Yes vs. No	40 vs. 19	0.31 (0.16–0.60)	0.001	0.38 (0.18–0.77)	0.007

Notes: ^†^ tested with Cox proportional hazards model. Abbreviations: HR, hazard ratio; CI, confidence interval; ECOG PS, ECOG performance status; ADC, adenocarcinoma; BM, brain metastasis; WT, wild type; MT, mutant type; PDC, patient-derived cell.

## Data Availability

Next-generation sequencing files of refractory lung cancer PDC samples have been deposited in the National Center for Biotechnology Information Gene Expression Omnibus (GSE165611 and GSE229535) for RNA-seq and Sequence Read Archive (PRJNA694788) for target-seq.

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
