# Peer review of "Drug Response of Patient-Derived Lung Cancer Cells Predicts Clinical Outcomes of Targeted Therapy"

_cancers, 2024, doi:10.3390/cancers16040778_

Round 1
Reviewer 1 Report
Comments and Suggestions for Authors
I would like to thank the authors for their very impressive work. Studying sensitivity in a short period of time such as 15 days is very important in terms of applicability in clinical practice.
The design of the study and the writing of the article were impressive.
I just have a few suggestions;
1. It is not necessary to give the result of the study in the last sentence of the introduction, you can omit the last sentence.
2. Table 1 shows that there are 19 patients without PDC response in patients treated with EGFR- or ALK-TKIs, and in line 255, it is stated that the clinical, pathological, and genomic characteristics analysis of 10 patients was performed. Were the other 9 patients not evaluated?
Author Response
We sincerely appreciate reviewer for the helpful reviews of our manuscript. We have
revised our manuscript extensively according to the reviewer’s valuable comments. We
hope we have addressed all the points to your satisfaction.
1. It is not necessary to give the result of the study in the last sentence of the introduction, you
can omit the last sentence.
Response:
We appreciated reviewer’s concerns. In line with the reviewer's suggestion, we have
omitted the last sentence of the introduction on page 2.
2. Table 1 shows that there are 19 patients without PDC response in patients treated with EGFRor ALK-TKIs, and in line 255, it is stated that the clinical, pathological, and genomic
characteristics analysis of 10 patients was performed. Were the other 9 patients not evaluated?
Response:
Only 10 patients (PDC non-response) were assessed for potential causes, as indicated in
the table S6. We regret that we were unable to collect sufficient information to assess
and draw definitive conclusions regarding the putative cause for nine PDC nonresponse patients.

Reviewer 2 Report
Comments and Suggestions for Authors
This study investigated intratumor heterogeneity's impact on responses to targeted therapies in lung cancer patients with identical driver oncogenes. Patient-derived cell (PDC)-based drug sensitivity tests were conducted on 397 PDCs from 487 lung cancers, revealing a correlation between standardized area under the curve (AUC) values and tumor response. Specifically, among chemo-naive EGFR/ALK-positive NSCLC patients, PDC responders exhibited superior response rates and progression-free survival compared to non-responders. The PDC-based drug sensitivity test proved valuable in predicting individualized clinical outcomes beyond genomic alterations for patients with EGFR- or ALK-positive NSCLC. The study is interesting and will interest the readers and enhance the clinical outcomes.
Minor Comments:
1. In the drug screening study, the authors did not explicitly mention the inclusion of antibodies against EGFR. Considering the significance of antibodies, especially against EGFR and AXL, in lung cancer treatment, the authors are encouraged to incorporate relevant information into the introduction or discuss it in comparison with their hypothesized regimens. Acknowledging studies involving EGFR and AXL antibodies or kinase inhibitors, either alone or in combination with other agents, would provide a more comprehensive context and contribute to a nuanced understanding of potential therapeutic strategies for lung cancer patients.
Author Response
We sincerely appreciate reviewer for the helpful reviews of our manuscript. We have
revised our manuscript extensively according to the reviewer’s valuable comments. We
hope we have addressed all the points to your satisfaction.
1. In the drug screening study, the authors did not explicitly mention the inclusion of antibodies against EGFR. Considering the significance of antibodies, especially against EGFR and AXL, in lung cancer treatment, the authors are encouraged to incorporate relevant information into the introduction or discuss it in comparison with their hypothesized regimens. Acknowledging studies involving EGFR and AXL antibodies or kinase inhibitors, either alone or in
ombination with other agents, would provide a more comprehensive context and contribute to anuanced understanding of potential therapeutic strategies for lung cancer patients.
Response:
We appreciated reviewer’s concerns. We acknowledge the reviewer's perspective on the significance of antibodies in cancer treatment. However, our in vitro drug screening platform has limitations in evaluating the efficacy of antibodies. Initially, we designed the drug screening panel to include only small molecules. Nevertheless, we are open to enhancing the drug screening panel by incorporating antibodies and exploring combination treatments. We appreciated reviewer’s suggestions.
Reviewer 3 Report
Comments and Suggestions for Authors
This work aims to test the capability of a novel drug screening method to direct medical workers to an optimal anti cancerous drug to a given patients, this based on the sensitivity of cancer cells to a given drug.
Overall, this work had very high medical and scientific valued thus I recommend its publication.
The introduction can be expended to include basic information on the drugs used in this work and on the role of genetic profiling of tumors. More over, the type of study should be stated clearly- control trial with case studies reports.
Author Response
We sincerely appreciate reviewer for the helpful reviews of our manuscript. We have revised our manuscript extensively according to the reviewer’s valuable comments. We hope we have addressed all the points to your satisfaction.
The introduction can be expended to include basic information on the drugs used in this work and on the role of genetic profiling of tumors. More over, the type of study should be stated clearly- control trial with case studies reports.
Response:
We appreciated reviewer’s concerns. We have inserted statements about basic
information in introduction section on page 2 as below and added references in
reference section.
few decades—of several drugs targeting unique oncogenic driver mutations [2]. For example, an EGFR-targeting drug is the first molecularly targeted therapy that has led to a paradigm shift in the management of patients with advanced nonsmall cell lung cancer (NSCLC) [3]. This drug has been used as a standard firstline treatment for patients with lung cancer harboring EGFR exon 19 deletion or exon 21 L858R mutations from multiple randomized clinical trials [4-8]. Molecular testing for patients newly diagnosed with
